# Tocotrienol in Pre-Eclampsia Prevention: A Mechanistic Analysis in Relation to the Pathophysiological Framework

**DOI:** 10.3390/cells11040614

**Published:** 2022-02-10

**Authors:** Zaleha Abdullah Mahdy, Kok-Yong Chin, Nik Lah Nik-Ahmad-Zuky, Aida Kalok, Rahana Abdul Rahman

**Affiliations:** 1Department of Obstetrics and Gynaecology, Universiti Kebangsaan Malaysia Medical Centre, Cheras, Kuala Lumpur 56000, Malaysia; aidahani.mohdkalok@ppukm.ukm.edu.my (A.K.); drrahana@ukm.edu.my (R.A.R.); 2Department of Pharmacology, Faculty of Medicine, Universiti Kebangsaan Malaysia, Cheras, Kuala Lumpur 56000, Malaysia; 3Department of Obstetrics and Gynaecology, School of Medical Sciences, Universiti Sains Malaysia, Kubang Kerian 16150, Malaysia; nikzuky@usm.my

**Keywords:** anti-inflammation, antioxidant, hypertension, ischaemia, vitamin E

## Abstract

The pathophysiology of pre-eclampsia involves two major pathways, namely systemic oxidative stress and subsequent generalised inflammatory response, which eventually culminates in endothelial cell injury and the syndrome of pre-eclampsia with multi-organ dysfunction. Aspirin has been used to reduce the risk of pre-eclampsia, but it only possesses anti-inflammatory properties without any antioxidant effect. Hence, it can only partially alleviate the problem. Tocotrienols are a unique form of vitamin E with strong antioxidant and anti-inflammatory properties that can be exploited as a preventive agent for pre-eclampsia. Many preclinical models showed that tocotrienol can also prevent hypertension and ischaemic/reperfusion injury, which are the two main features in pre-eclampsia. This review explores the mechanism of action of tocotrienol in relation to the pathophysiology of pre-eclampsia. In conclusion, the study provides sufficient justification for the establishment of a large clinical trial to thoroughly assess the capability of tocotrienol in preventing pre-eclampsia.

## 1. Introduction

Pre-eclampsia is a clinical syndrome characterised by raised blood pressure and multi-organ dysfunction during the second half of pregnancy, labour or the puerperium in a previously normotensive woman [1]. It is the leading cause of maternal and perinatal mortality and morbidity [2]. The global incidence of pre-eclampsia has increased from 16.30 million in 1990 to 18.08 million in 2019 (a 17.80% increase) despite global improvement in maternal health [3]. Chronic diseases, such as diabetes mellitus, obesity, chronic hypertension, renal disease, and connective tissue disorders, are risk factors of pre-eclampsia [4].

Pre-eclampsia tends to recur, but a window of opportunity for its screening and prevention exists. The current management of pre-eclampsia involves screening (by risk factors, imaging, or biomarkers), counselling, continuous and meticulous monitoring and pharmacological control of blood pressure before and after delivery, as well as the prophylactic use of low-dose aspirin in high-risk women, magnesium sulphate to prevent eclampsia, and betamethasone to promote fetal lung maturity in pregnancies <34 weeks [5]. Mapping the pathophysiology of a disease as accurately as possible is necessary to justify the use of the preventive measure and institute prevention by pharmacoprophylaxis. Over the years, many candidate drugs have been clinically tested, the most established being aspirin and calcium [6]. Low-dose aspirin (81 mg/day) has been recommended by the US Preventive Task Force as the preventive drug, because it reduces pre-eclampsia risk and has multiple benefits on maternal and neonatal outcomes [7]. However, a recent meta-analysis revealed that aspirin at doses lower than 150 mg/day could not remarkably prevent preterm pre-eclampsia [8]. In addition, low-dose aspirin may be contraindicated in patients with hypersensitivity to non-steroidal anti-inflammatory drugs and gastrointestinal and genitourinary bleeding tendencies [9]. Aspirin might also cross the placenta to the foetus and pose a theoretical risk of foetal intracranial haemorrhage despite the lack of epidemiological evidence [10]. For calcium supplementation, the risk reduction is only about 20%, and is higher amongst populations with low calcium intake [11]. Recent studies have demonstrated the potential of biologics, such as recombinant vascular endothelial growth factor (VEGF) and placental growth factor, in managing pre-eclampsia, but they are still in the development phase [12]. Currently, the pursuit of an alternative agent for pre-eclampsia prevention is still ongoing.

Oxidative stress has been associated with pre-eclampsia in many observational studies [13,14]. Thus, many researchers have advocated for the potential role of antioxidants in preventing pre-eclampsia [15]. The antioxidants that have been actively investigated include coenzyme Q10 [16], melatonin [17,18], and curcumin [19]. However, meta-analyses have reported that antioxidants are ineffective in pre-eclampsia prevention, and the effects on foetal outcomes are heterogeneous [20,21]. Results of trials involving vitamin E, particularly α-tocopherol, have been hugely disappointing [22]. However, the effects of another form of vitamin E, tocotrienol, in pre-eclampsia prevention have been less explored. Only one study examined the effects of tocotrienol-rich fraction (TRF) derived from palm oil with promising outcomes [23]. In this study, TRF reduced the risk of pre-eclampsia by 64% [23]. Before embarking on expanding such clinical trials further, scrutinising the currently accepted pathophysiology of pre-eclampsia and the potential targets of tocotrienol in general would be prudent. Therefore, through this review, we set out to critically evaluate the scientific merits of tocotrienol as a preventive agent of pre-eclampsia.

## 2. Current Concept on the Pathophysiology of Pre-Eclampsia

The pathophysiology of pre-eclampsia has evolved progressively over the years, and the most recent concept embraces oxidative stress and the inflammatory process [24]. The pathogenesis of pre-eclampsia involves maternal and placental pathways. The source of insult originates from maternal biochemical anomalies, due to chronic diseases or placental developmental anomalies in the form of restricted spiral artery invasion by foetal trophoblasts [25]. Maternal origins of insult may arise from pre-existing diseases, such as diabetes mellitus, chronic hypertension, and obesity, whereas placental anomalies may be the result of immunological incompatibility between maternal and paternal human leukocyte antigens. Maternal metabolic diseases like ischaemia/reperfusion of the placental bed releases biochemical signals into the maternal circulation. These signals converge at the endothelium and form the final common pathway that results in pre-eclampsia, which is marked by endothelial dysfunction, raised blood pressure, and potential multi-organ complications. The humoral signals that produce pre-eclampsia and endothelial dysfunction consist of oxidative and inflammatory agents [12,25].

In the early stage of pregnancy, oxygen concentration within the trophoblast is maintained at a relatively low level to avoid teratogenic effects on the foetus. The hypoxic condition also up-regulates hypoxia-inducible factor 1 alpha, which, in turn, increases endothelial nitric oxide (NO) synthase to produce NO radicals that facilitate trophoblast proliferation and invasion. Towards the end of the first trimester, uteroplacental circulation is developed to allow a higher concentration of oxygen to reach the embryo to facilitate cell differentiation and foetal development. Oxidative stress develops easily at the syncytiotrophoblast, which is a special epithelium necessary for the transport of solute to the foetus and the production of hormones. It has a membrane rich in unsaturated fatty acid, but low in antioxidant enzymes; thus, it is susceptible to oxidative damage (summarised in [26]).

Defective placental implantation onto the uterine wall impairs uterine spiral arteriole remodelling by extravillous trophoblasts, which subsequently reduces placental perfusion. The resultant cycles of placental ischaemia and reperfusion produce oxidative stress [27]. Notably, the hypoxia/reperfusion cycles activate xanthine oxidase, nicotinamide adenine dinucleotide phosphate oxidase, and electron transport chain (complexes I and III) in the mitochondria to generate oxygen radicals. These radicals are reduced to hydrogen peroxide by manganese, copper, and zinc superoxide dismutase in the intermembranous space of the mitochondria, and subsequently reduced to water by glutathione peroxidase or catalase [24]. Maternal metabolic derangements could result in impaired antioxidant defence and higher oxidants, which allow these radical oxygen species to cause cellular and tissue damage. Oxidative stress can also activate the nuclear factor kappa-B (NF-κB) pathway, which mediates the inflammation process [24]. Damaged tissues release cell debris antiangiogenic factors, such as soluble fms-like tyrosine kinase-1 (sFlt-1), soluble endoglin, and cytokines [27]. sFlt-1 binds to VEGF and reduces its bioavailability to maternal endothelial cells, which results in the reduction of NO production and triggers vasoconstriction [28]. It also sensitises human umbilical artery endothelial cells to the action of pro-inflammatory cytokines [29]. The hypoxic condition up-regulates p53 expression (indicative of pro-apoptotic signalling) and down-regulates Bcl-2 expression (indicative of anti-apoptotic signalling), which further escalates the apoptosis of syncytiotrophoblasts [30,31]. Oxidative stress also activates Wnt signalling and matrix metalloproteinases (MMPs), and thus promotes the invasiveness of trophoblasts and increases the risk of pre-eclampsia [32].

Uterine immune cells are a major regulator of inflammation. During placentation, various immune cells, such as macrophages, natural killer cells, T-cells, B-cells, and dendritic cells, guide the proper invasion of trophoblasts [33]. In pre-eclampsia, chronic placental and peripheral inflammation, marked by increased tumour necrosis factor-α (TNFα) and interleukin (IL)-6 and decreased IL-10 and IL-4, are observed [34]. In particular, TNFα triggers endothelial dysfunction by reducing NO production, which activates the NF-κB pathway, propagates inflammation, increases endothelin-1 (ET-1), and causes vasoconstriction [35]. TNFα, IL-6, and IL-17 also activate B-cells to secrete agonistic autoantibody against angiotensin II type-1 (AT-1) receptor, which, upon binding, causes the release of proinflammatory cytokines as well as ET-1 and sFlt-1 in the vasculature, and subsequently causes vasoconstriction [36]. Overall, the pathophysiology of pre-eclampsia is summarised in Figure 1.

## 3. Vitamin E: Tocopherol and Tocotrienol

Although α-tocopherol was the first form of vitamin E to be recognised, natural vitamin E has eight lipophilic forms, including four tocopherols (α, β, γ and δ) and four tocotrienols (α, β, γ and δ) [37]. Tocopherols are saturated forms of vitamin E, whereas tocotrienols are the unsaturated forms [38]. α-Tocopherol is more ubiquitous in nature, whereas tocotrienol can be found in oil palm kernel, annatto bean, and rice bran. Tocotrienol isomers are usually found in mixtures of varying compositions in natural sources [39]. For instance, palm-derived TRF consists of a mixture of ≈75% tocotrienol (α, β, γ and δ) and ≈25% α-tocopherol, whereas annatto tocotrienol consists of ≈90% δ-tocotrienol and ≈10% γ-tocotrienol [40]. The effect of α-tocopherol on the treatment efficacy of tocotrienol mixtures is a subject of debate [41], because α-tocopherol competes with other vitamin E isomers to bind to a-tocopherol transfer protein (ATTP) in the liver before being released into the circulation [42]. α-Tocopherol preferentially binds with ATTP; thus, it has a higher bioavailability than other vitamin E isomers; the lower bioavailability might hinder the biological activities of the isomers. However, a comparative study did not find any remarkable difference in the skeletal and metabolic activities of palm TRF (containing α-tocopherol) and annatto tocotrienol (lacking α-tocopherol) in rats [43,44].

Vitamin E has been actively investigated for its effects in managing pre-eclampsia, because it could tackle several pathophysiological pathways of the disease, theoretically. Firstly, vitamin E is a well-established antioxidant [45,46] that can suppress the oxidative damage caused by repeated placental ischaemia/reperfusion. Secondly, vitamin E and its metabolites are anti-inflammatory agents [47,48,49] that could suppress systemic inflammation caused by the release of damage-associated molecules. Thirdly, vitamin E supplementation may regulate blood pressure, but the evidence is debatable [50].

Until recently, most studies have focused on α-tocopherol, which is the predominant form of vitamin E in body tissues. A previous Cochrane review demonstrated that vitamin E (α-tocopherol) combined with other supplements had no effect on pre-eclampsia [51]. This result was because most women have sufficient vitamin E from their diet, and a short-term supplementation is unlikely to have an effect on pregnant women. In addition, most comparative studies showed that α-tocopherol is less efficacious than tocotrienols in managing chronic diseases [40,52,53]. These limitations justify the shift of attention from α-tocopherol to tocotrienol in pre-eclampsia management.

## 4. Antioxidant Effects of Tocotrienol

Tocopherols and tocotrienols share similar structures, such as an electron-rich chromanol ring and a long carbon tail. The carbon tail of tocopherols is saturated, whereas the tail of tocotrienols contains three double bonds (Figure 2) [54]. The structural difference between tocopherols and tocotrienols dictates their antioxidant potentials on a lipid membrane. Compared with α-tocopherol, α-tocotrienol has a stronger ability to disorganise lipid bilayers and is distributed more homogeneously within the lipid bilayers. α-Tocotrienol is also closer to the surface of the membrane, which facilitates its interaction with free radicals and the redox recycling process [46,55].

Nuclear factor erythroid 2-related factor 2 (NRF2) is a master regulator of cellular antioxidants, as it governs the expression of various antioxidants in response to oxidative stress [56]. Palm TRF up-regulates NRF2 expression, which increases the nuclear translocation of NRF2 in mouse liver [57]. Moreover, δ-tocotrienol can stabilise the NRF2 activation induced by 5-fluorouracil in human oral keratinocytes, which can cause a sustained increase in the expression of heme oxygenase-1 and NAD(P)H:quinone oxidoreductase-1 [58]. Another study on the antioxidant effects of δ-tocotrienol osteoblasts found that the activation of the NRF2 and PI3K/AKT pathways is necessary for the protection of δ-tocotrienol against oxidative stress [59].

The antioxidant effects of tocotrienol, through direct electron scavenging activities or the activation of the NRF2 system, are translated in vivo. Tocotrienol isomers or mixtures exert antioxidant effects in various disease models, such as osteoporosis [60,61,62], gastric ulcers [63], cardiovascular diseases [64,65], and metabolic syndrome [44]. Interestingly, a recent clinical trial demonstrated the more pronounced antioxidant benefits of TRF amongst women than men [66]. A recent meta-analysis of human clinical trials showed that tocotrienol supplementation can improve the redox status of patients, as evidenced by the markedly reduced levels of lipid peroxidation products [67]. Thus, the antioxidant effects of tocotrienol can be exploited to prevent the oxidative stress caused by placental ischaemia/reperfusion injury in pre-eclampsia.

## 5. Anti-Inflammatory Effects of Tocotrienol

The NF-κB pathway is an important regulatory pathway for inflammation [68]. Previous studies showed that γ- and δ-tocotrienols inhibit the TNFα-induced activation of NF-κB by increasing intracellular dihydroceramides (a type of sphingolipid), cellular stress, and A20 [69,70]. A20 is a negative regulator of the NF-κB pathway by acting as a deubiquitinase [71]. γ-Tocotrienol suppresses the phosphorylation and degradation of TNF-induced NF-κB inhibitor-α (IκBα) by preventing IκBα kinase activation and thus, abolishes the phosphorylation and nuclear translocation of p65 and the transcription of NF-κB-dependent reporter genes [72]. This mechanism, together with the suppression of CCAAT/enhancer-binding protein beta, explains the suppressive effects of γ-tocotrienol on the expression of IL-6 and granulocyte colony-stimulating factor in primary bone marrow-derived macrophages, including chemokine response in the pathogenesis of asthma [70].

NF-κB is also indispensable for the expression of cyclooxygenase, which is responsible for converting arachidonic acid to prostaglandin precursors [73]. δ-Tocotrienol selectively decreases the expression of cyclo-oxygenase (COX)-2 and 5-lipoxygenase, but not COX-1, in lipopolysaccharide-induced microglial cells [74]. This decrease is accompanied by a reduction in the expression of inducible NO synthase and IL-1β [74]. The anticancer activities of γ-tocotrienol are also attributed to its ability to suppress the expression of COX-2, MMP-2, and MMP-9 proteins [75]. A comparison of the anti-inflammatory effects of tocotrienol isomers and RAW264.7 macrophages showed that only α-tocotrienol can remarkably suppress TNFα expression [76]. Besides, all tocotrienol isomers can reduce the expression of IL-6 and COX-2, but γ-tocotrienol is less efficacious in decreasing prostaglandin E_2_ release. The COX-down-regulating activities of tocotrienol isomers are not shared by α-tocopherol [76]. Another study on the anti-asthma potential of vitamin E reported that γ-tocotrienol is more effective than α-, γ-, and δ-tocopherols in suppressing IL-13-stimulated eotaxin-3 production in human lung epithelial A549 cells [77]. These studies highlighted that the anti-inflammatory effects of each tocotrienol isomer against each cytokine could be different, and a mixture with the full spectrum of tocotrienols, such as palm TRF, could be more effective in suppressing inflammation.

The anti-inflammatory effects of tocotrienol mixtures can be explored with the aim of suppressing the inflammation caused by maternal metabolic conditions or inflammation in pre-eclampsia. Besides, animal studies also suggested that tocotrienols could prevent aspirin-induced gastric lesions by limiting lipid peroxidation [78]. Tocopherols and tocotrienols have gastroprotective effects against oxidative stress, but only tocotrienols can block stress-induced changes in gastric acidity and gastrin level [79]. Palm TRF increases the level of the constituently expressed COX-1 enzyme and prostaglandin E_2_ to exert its gastric protective effects [80]. γ-Tocopherol prevents, whereas α-tocopherol worsens, aspirin-induced gastric injury [81]. Considering these studies, tocotrienols could mitigate the gastrointestinal bleeding side effects of aspirin. Nonetheless, a recent meta-analysis of human clinical trials showed that the effects of tocotrienol supplementation on circulating inflammatory mediators were not consistent [67].

## 6. Role of Tocotrienol in Regulating Blood Pressure

Hypertension is one of the main features of pre-eclampsia; hence, the effects of tocotrienol in regulating blood pressure deserve a closer look. Numerous preclinical studies reported the blood pressure-regulating mechanism of tocotrienol. Newaz et al. [82] supplemented male spontaneously hypertensive rats (SHR) with γ-tocotrienol (15, 30 and 150 mg/kg diet) for 3 months and found that γ-tocotrienol reduced systolic blood pressure at all doses. This beneficial effect was accompanied by a reduction in circulating malondialdehyde and nitrite levels and an increase in NO synthase activity in the blood vessels [82]. Muharis et al. [83] showed that the exposure of the aortic ring of SHR to palm TRF enhanced acetylcholine, but not sodium nitroprusside-induced relaxation [83]. α-Tocopherol has similar effects as TRF. However, this study did not pinpoint any specific mechanism of tocotrienol.

In diet-induced metabolic syndrome models, palm TRF and annatto tocotrienol supplementation (60 and 100 mg/kg body weight, respectively) for 8 weeks decreased the blood pressure and other metabolic derangements in rats, such as hyperglycemia and dyslipidemia. These changes are associated with the reduction of circulating IL-6 and IL-1α, but not TNFα [44,84]. Other researchers proposed that palm TRF could exert antihypertensive effects at a lower dose (60 mg/kg) and a shorter period (4 weeks) in rats fed with high-fat diet [85]. The effectiveness of individual vitamin E isomers (α-tocopherol, α-, γ-, and δ-tocotrienols; each at 85 mg/kg) against metabolic syndrome induced by high-fat high-carbohydrate diet in rats was also examined. Only γ- and δ-tocotrienols can normalise the blood pressure of rats with metabolic syndrome [86].

Hyperhomocysteinemia is a classical risk factor of hypertension that can lead to endothelial cell damage and reduced blood vessel flexibility [87]. Palm TRF supplementation (60 and 150 mg/kg diet) for 5 weeks in rats fed with high-methionine diet decreased the levels liver, heart, and aortic lipid peroxidation products, increased heart glutathione peroxidase and circulating NO, but did not change the systolic pressure of the animals [88,89,90]. The ineffectiveness of tocotrienol in normalising blood pressure in hyperhomocysteinemia could be due to the more extensive damage on endothelial tissues, but further studies are needed.

The effects of self-emulsified palm TRF on blood pressure and arterial stiffness were also tested in a randomised controlled trial. Rasool et al. reported that self-emulsified palm TRF (50, 100, and 200 mg) supplementation in healthy men for 2 months did not modify their blood pressure and lipid profile, but remarkably reduced pulse wave velocity and augmentation index. The lack of effects of palm TRF on these subjects is expected because they were not hypertensive [91]. By contrast, Aminuddin et al. demonstrated that palm TRF at 100 mg daily from 12–16 gestational weeks to delivery substantially reduced the risk of pregnancy-induced hypertension (odds ratio [OR]: 0.254; 95% confidence interval [CI]: 0.07–0.93) and pre-eclampsia (OR: 0.030, 95% CI: 0.001–0.65) amongst pregnant women [23]. However, the CI values for both effects were very large, which implies that the effect might not be consistent for all patients.

Based on these studies, tocotrienol might be able to prevent hypertension by normalising the circulating NO level and reducing inflammation and oxidative stress. By extension, these protective effects suggest that tocotrienol has a protective role in suppressing hypertension caused by pre-eclampsia, because the principal mechanisms of blood pressure regulation in various health conditions are similar. However, most of the hypotensive studies were conducted in rats, which have a different metabolic rate and life span compared with humans [92]. Most of the studies summarised were performed in adult rats. Cheng et al. [85] demonstrated that at least 4 weeks of treatment is necessary for TRF to reverse hypertension. Approximately 10.5 days in rats are equivalent to 1 year in humans at adulthood [92], which means that humans need to consume tocotrienol for at least 2.67 years to effectively reverse hypertension. Given this limitation, a direct translation of the effects of tocotrienol on blood pressure from existing animal models may be problematic because it is not tested in a pre-eclampsia model. Preliminary data from Aminuddin et al. [23] suggested that tocotrienol could be used as a preventive agent rather than a reversal agent, based on animal studies. More comprehensive human clinical trials with larger sample size are required to prove the effects of TRF in preventing hypertension amongst women with pre-eclampsia.

## 7. Role of Tocotrienol in Protecting against Ischaemic Injury

Placental ischaemic injury is a major feature of pre-eclampsia. Thus, the protective effects of tocotrienol against ischaemic injury would be of interest. An investigation of the human placentome revealed that γ-tocopherol is preferentially deposited at the basal plate but not in the human umbilical cord, whereas β-tocopherol is found at both sites. The levels of β- and γ-tocopherols are higher than that of α-tocopherol. Moreover, β- and γ-tocotrienols are found in minute amounts in the placentome [93]. These observations showed that vitamin E isomers (tocotrienols and tocopherols) are deposited at the placentome and probably have a functional role as parts of the non-enzymatic antioxidant defence system.

Although the effects of tocotrienol on placental ischaemia have not been examined, they have been investigated in other organs, particularly in the brain and heart. Palm TRF (100 mg twice daily) supplementation in mongrel canines for 10 weeks before transient middle cerebral artery occlusion was able to attenuate the ischaemic lesion and retained white matter fibre tract connectivity by improving the circulation at the stroke area. Palm TRF also stimulated the expression of arteriogenic markers and suppressed MMP-2 activity at the stroke area [94]. Mice supplemented with commercialised palm TRF (Tocovid, 200 mg/kg/day for 1 month) and subjected to transient middle cerebral artery occlusion showed reduced brain lesion, oxidative damage, and autophagy/apoptosis markers, as well as increased NRF2 pathway activation and improved function [95]. Mishima et al. differentiated the individual protective effects of vitamin E isomers and reported that α-tocotrienol, α-tocopherol, and γ-tocopherol are more effective than γ-tocotrienol, δ-tocopherol, and δ-tocotrienol in reducing brain infarct lesions after middle cerebral artery occlusion in mice [96]. This finding was further confirmed by another study, which reported that α-tocotrienol protected against ischaemic cerebral injury via the up-regulation of multidrug resistance-associated protein 1 and the down-regulation of its modulator miR-199a-5p [97].

For ischaemic heart disease, the supplementation of male rats with red palm oil (7 g/kg diet for 6 weeks) rich in vitamin E (including tocotrienol), B-carotenes, and various fatty acids preserved cardiac output, cGMP production, and total polyunsaturated fatty acid content of the rodent heart subjected to ischaemic/reperfusion injury [98]. cGMP production is triggered by NO [99]; thus, red palm oil could increase NO production by NO synthase, but this mechanism was not examined in the study. Das et al. [100] compared the protective effects of α-, β-, γ-, and δ-tocotrienol isomers (20 mol/kg/day for 4 weeks, after 4 weeks on 2% cholesterol diet) on excised proline heart using an ischaemic/reperfusion model. α and γ-Tocotrienols suppress the expression of genes modulating ET-1, MMP-2, MMP9, and thyroid hormone-responsive protein Spot14, and increased TGFβ [100].

Although direct evidence is lacking, the examples aforementioned highlight several targets of placental ischaemia/reperfusion injury that are modulated by TRF. These targets include NRF2, which modulates cellular antioxidant response [101]; NO/cGMP pathway, which plays a major role in vasoconstriction [99]; and MMPs, which are important for vascular and uterine remodelling [99]. TRF could suppress MMP production by extravillous trophoblasts and decidual cells at the early stage of pregnancy, which contradicts its other beneficial effects in preventing pre-eclampsia. This aspect should be investigated in a valid preclinical model of pre-eclampsia.

## 8. Safety Profile of Palm TRF amongst Pregnant Women

Other important aspects concerning the use of palm TRF amongst pregnant women are bleeding risk and reproductive toxicity. Vitamin E supplementation (400 IU tocopherol for 8 years) has been associated with increased risk of haemorrhagic stroke amongst male physicians in the Physicians’ Health Study II [102], but this effect was not shown in the Women’s Health Study (600 IU tocopherol for 10 years) [103]. Real-world safety data for tocotrienol is lacking. A subacute toxicity study showed that palm TRF above 500 mg/kg body weight can prolong bleeding and clotting time in female mice [104]. Based on the body surface conversion formula [105], this dose is equivalent to 2400 mg for a human adult weighing 60 kg; hence, the daily dose used in the clinical trial on pre-eclampsia (100 mg) was much lower. With regard to the concurrent use of tocotrienol and other anticoagulants, a recently published randomised trial reported that the concurrent use of aspirin, clopidogrel, and palm TRF could reduce the incidence of aspirin resistance [106]. Only one study on the female reproductive toxicity of palm TRF was reported. Supplementation of palm TRF (200, 500, and 1000 mg/kg) 2 weeks before pregnancy and continuing throughout the pregnancy (for about 21 days) did not result in a remarkable change in pup number, body weight, and body length [107]. However, no further assessment was done on the dams or pups.

A recent study highlighted that α-tocotrienol could reduce the viability, growth, migration, epithelial–mesenchymal transition, invasion, and angiogenesis of trophoblasts in vitro. These effects were mediated by the down-regulation of miR-429 and the up-regulation of zinc finger E-box-binding homeobox (ZEB). The same study revealed that placenta sampled from pre-eclamptic women expressed higher miR-429 and lower ZEB levels [108]. These findings imply that δ-tocotrienol could be toxic to trophoblasts. Given these observations, we postulated that a natural mixture with lower δ-tocotrienol could be safer for embryos. Compared with annatto tocotrienol with ≈90% δ-tocotrienol in the mixture, palm TRF usually contains less than 20% of δ-tocotrienol, which makes palm TRF a safer supplementation [40]. However, this speculation requires further validation from preclinical studies.

## 9. Future Research

Only one clinical trial has looked at a tocotrienol-predominant preparation in preventing pre-eclampsia [23]. In this randomised double-blind placebo-controlled trial involving 299 primigravidae without any other risk factors for pre-eclampsia, the risk of pre-eclampsia was remarkably reduced by 83%. Vitamin E was prescribed in the form of TRF, at a dose of 100 mg daily. Repeating this trial on a population with additional risk factors for pre-eclampsia, such as a history of pre-eclampsia and current diabetes mellitus, chronic hypertension, or other medical disorders that increase the risk of developing pre-eclampsia, would be of interest.

As prophylactic agents against pre-eclampsia, the combined use of aspirin and tocotrienols should be considered and investigated, as tocotrienols seem to be protective against aspirin-induced gastric lesions and, therefore, add a potential benefit besides affording antioxidant activity. Whether tocotrienols will enhance the anti-inflammatory property of aspirin is another pertinent research question. Embarking on an in vitro or animal study may be helpful to scrutinise this hypothesis before conducting a clinical trial combining tocotrienols and aspirin for pre-eclampsia prevention. Which preparation of vitamin E should be used in this endeavour is another question to ponder. Perhaps TRF, with its unique combination of tocotrienols and tocopherol, can be considered, given the abundance of supporting evidence of the benefit of TRF from studies on non-communicable diseases [39].

## 10. Conclusions

The pathophysiology of pre-eclampsia involves oxidative stress and inflammation activation. Anti-inflammatory agents, such as aspirin, only act upon a part of the disease mechanism, which results in a partial remedy in the preventive process. Tocotrienols are forms of vitamin E that possess potent antioxidant and anti-inflammatory characteristics; hence, they are potentially more appropriate preventive agents to combat pre-eclampsia, compared with tocopherol. Tocotrienols might also prevent hypertension and ischaemia/reperfusion injury in the placenta (Figure 3). Notably, the effects of tocotrienol on hypertension are contentious, because a long duration of treatment is required. Most of the mechanisms proposed are theoretical, because they are not proven yet in experimental models of pre-eclampsia. This limitation necessitates more fundamental studies on this topic. To date, only a single-centre small-scale clinical trial investigated the effects of TRF on pregnant mothers. Therefore, a large randomised clinical trial on a population of pregnant women at high risk of pre-eclampsia is warranted to shed more light on this matter.

## Figures and Tables

**Figure 1 cells-11-00614-f001:**
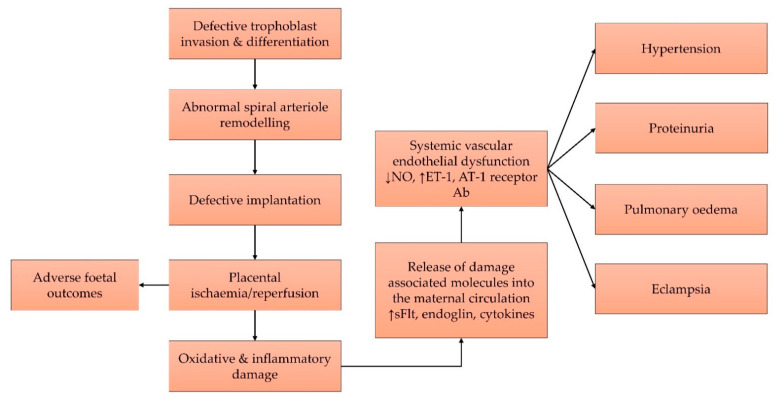
Pre-eclampsia begins with defective placental implantation, which leads to recurrent ischaemia and reperfusion cycles that result in oxidative and inflammatory damages to the tissue. Damage-associated molecules cause systemic endothelial dysfunction and various features of pre-eclampsia. Abbreviations: Ab, antibody; AT-1, angiotensin-1; ET-1, endothelin-1; NO, nitric oxide; sFlt-1, soluble fms-like tyrosine kinase 1.

**Figure 2 cells-11-00614-f002:**
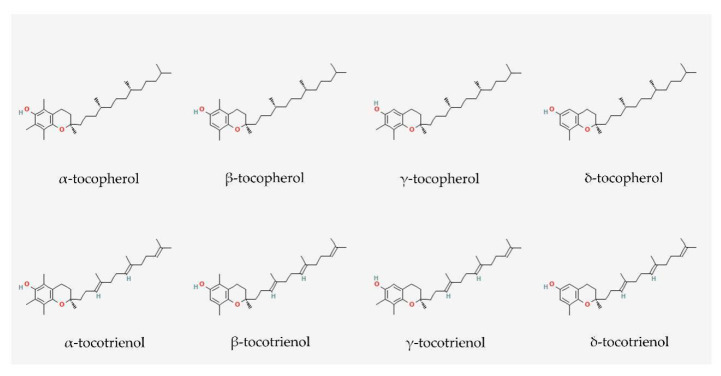
Molecular structures of tocopherols and tocotrienols. They consist of a chromanol ring and a long carbon tail. Tocopherols and tocotrienols have saturated and unsaturated carbon tails, respectively. Each isomer is distinct from each other by the position of the methyl side chain on the chromanol ring. The structures were obtained from PubChem.

**Figure 3 cells-11-00614-f003:**
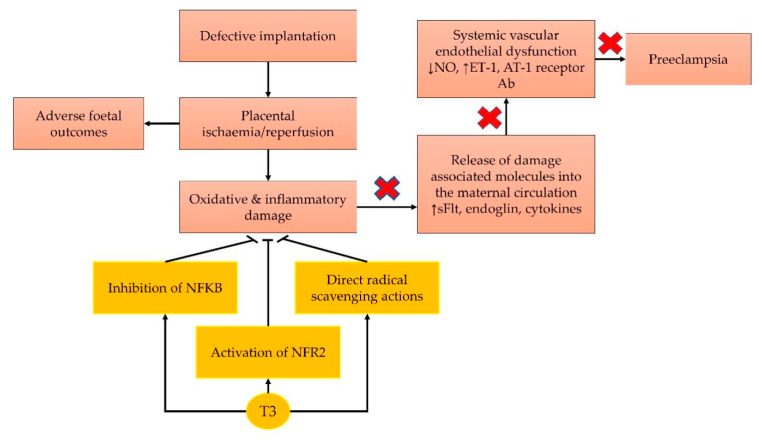
Theoretical framework of how tocotrienol prevents pre-eclampsia. Tocotrienol exerts anti-inflammatory and antioxidant actions, which prevent the oxidative and inflammatory damages caused by ischaemia. The same properties also prevent hypertension in pre-eclampsia. Abbreviation: Ab, antibody; AT-1, angiotensin-1; ET-1, endothelin-1; NF-κB, nuclear factor kappa-B; NO, nitric oxide; NRF2; nuclear factor erythroid 2-related factor 2; T3, tocotrienol; sFlt-1, soluble fms-like tyrosine kinase 1.

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
