# Peer review of "Tocotrienol in Pre-Eclampsia Prevention: A Mechanistic Analysis in Relation to the Pathophysiological Framework"

_cells, 2022, doi:10.3390/cells11040614_

Round 1

Reviewer 1 Report

The article is about an important subject.

A few specific concerns:

The data presented in the paper are known.

The conclusion is too general

Author Response

Thank you for reviewing our manuscript. We appreciate your comments and have responded in the attached response sheet. 

We look forward to receiving a favourable reply from you.

Reviewer 2 Report

This review paper decribes the role of tocotrienol (an active form of vit E) on preeclampsia from a mechanistic standpoint. Based on its antioxidant and anti-inflammatory actions, authors conclude that tocotrienol is able to prevent hypertension and ischemic/reperfusion injury, two main features in underlying preeclampsia.

This is a timely and well-written paper.

Page 2, I suggest the authors to describe better the functional antioxidants which are potential to treat preeclampsia (see the following articles to strengthen the sentence - PMID: 29766570, PMID: 31906255, PMID: 32916282, PMID: 33080891).

The topic of #3 on vitamin E, authors could describe general protective effects of vit E on preeclampsia and mention what is the advantages of tocotrienol compared to tocopherol.

Author Response

(The authors gave the same response as above.)

Reviewer 3 Report

In their paper titled " Tocotrienol in Prevention of Preeclampsia: A Mechanistic Analysis in Relation to the Pathophysiological Framework", Mahdy et al. describe that the preventable role of tocotrienol on the development of preeclampsia through the inhibition of inflammatory reaction. The authors reviewed the many kinds of effects of tocotrienol on human disorders and concluded the favorable effect on the pathogenesis of preeclampsia. This is an interesting manuscript and valuable for many clinicians. Also, the aim is sufficiently original. However, the overall quality of the manuscript is impaired by some problems including the quality of the evidences and the wrong evaluation. Furthermore, the overall quality of the manuscript would be improved by a thorough revision of the overall course. It is thought that further proofreading is needed to improve the quality of the text. I have provided specific suggestions below.

Major concerns:

The authors noted that tocotrienol has a hypotensive effect, but several studies in rodents have shown that it takes a long time for this effect to occur, which suggests that the effect on blood pressure during pregnancy, which is relatively short, may not be as great. This is consistent with the fact that antioxidants have been ineffective in the prevention and treatment of hypertensive disorders of pregnancy in many studies.

In their article, the authors report that TRF acts to decrease MMPs, and one of the pathogenic mechanisms of preeclampsia is thought to be a decrease in MMP production by extravillous trophoblasts and decidual cells. Suppression of MMPs by TRF might promote the pathogenesis of preeclampsia.

A 2015 Cochrane review reported that vitamin E supplementation to pregnant women had no effect on preeclampsia. The daily intake of vitamin E in normal women is not low in general, and it is unlikely that a short-term overdose would have a positive effect on pregnant women. Furthermore, excess lipid-soluble vitamins can be harmful to the fetus.

(Rumbold A, et al. Vitamin E supplementation in pregnancy. Cochrane Database of Systematic Reviews 2015: 9. DOI: 10.1002/14651858.CD004069.pub3)

Looking at the results of cited reference 19, the χ2 test showed that there was no significant difference in the inhibitory effect of TRF on the development of PE. If this is the only one manuscript that reported the "preventive effects of Palm Oil Vitamin E on preeclampsia," its efficacy must be questioned.

Author Response

(The authors gave the same response as above.)

Round 2

Reviewer 3 Report

The authors replied on each comment sincerely and the replies were appropriate. The quality of papers submitted for consideration includes enough reader's interest and scientific quality. The given paper satisfies requirements for publication of this journal.